# ISA-VAE: INDEPENDENT SUBSPACE ANALYSIS WITH VARIATIONAL AUTOENCODERS

## ABSTRACT

Recent work has shown increased interest in using the Variational Autoencoder (VAE) framework to discover interpretable representations of data in an unsupervised way. These methods have focussed largely on modifying the variational cost function to achieve this goal. However, we show that methods like $\beta$-VAE simplify the tendency of variational inference to underfit causing pathological overpruning and over-orthogonalization of learned components. In this paper we take a complementary approach: to modify the probabilistic model to encourage structured latent variable representations to be discovered. Specifically, the standard VAE probabilistic model is unidentifiable: the likelihood of the parameters is invariant under rotations of the latent space. This means there is no pressure to identify each true factor of variation with a latent variable. We therefore employ a rich prior distribution, akin to the ICA model, that breaks the rotational symmetry. Extensive quantitative and qualitative experiments demonstrate that the proposed prior mitigates the trade-off introduced by modified cost functions like $\beta$-VAE and TCVAE between reconstruction loss and disentanglement. The proposed prior allows to improve these approaches with respect to both disentanglement and reconstruction quality significantly over the state of the art.

## 1 INTRODUCTION

Recently there has been an increased interest in unsupervised learning of disentangled representations. The term *disentangled* usually describes two main objectives: First, to identify each true factor of variation with a latent variable, and second, interpretability of these latent factors (Schmidhuber, 1992; Ridgeway, 2016; Achille & Soatto, 2017). Most of this recent work is inspired by the $\beta$-VAE concept introduced in Higgins et al. (2016), which proposes to re-weight the terms in the evidence lower bound (ELBO) objective. In Higgins et al. (2016) a higher weight for the Kullback-Leibler divergence (KL) between approximate posterior and prior is proposed, and putative mechanistic explanations for the effects of this modification are studied in Burgess et al. (2017); Chen et al. (2018). An alternative decomposition of the ELBO leads to the recent variant of $\beta$-VAE called $\beta$-TCVAE (Chen et al., 2018), which shows the highest scores on recent disentanglement benchmarks.

These modifications of the evidence lower bound however lead to a trade-off between disentanglement and reconstruction loss and therefore the quality of the learned model. This trade-off is directly encoded in the modified objective: by increasing the $\beta$-weight of the KL-term, the relative weight of the reconstruction loss term is more and more decreased. Therefore, optimization of the modified ELBO will lead to latent encodings which have a lower KL-divergence from the prior, but at the same time lead to a higher reconstruction loss. Furthermore, we discuss in section 2.4 that using a higher weight for the KL-term amplifies existing biases of variational inference, potentially to a catastrophic extent.

There is a foundational contradiction in many approaches to disentangling deep generative models (DGMs): the standard model employed is not identifiable as it employs a standard normal prior which then undergoes a linear transformation. Any rotation of the latent space can be absorbed into the linear transform and is therefore statistically indistinguishable. If interpretability is desired, the modelling choices are setting us up to fail.

We make the following contributions:

- We show that the current state of the art approaches employ a trade-off between reconstruction loss and disentanglement of the latent representation.

- In section 2.3 we show that variational inference techniques are biased: the estimated components are biased towards having orthogonal effects on the data and the number of components is underestimated.

- We provide a novel description of the origin of disentanglement in $\beta$-VAE and demonstrate in section 2.4 that increasing the weight of the KL term increases the over-pruning bias of variational inference.

- To mitigate these drawbacks of existing approaches, we propose a family of rotationally asymmetric distributions for the latent prior, which removes the rotational ambiguity from the model. This approach resembles independent component analysis (ICA) for variational autoencoders.

- We propose to use a prior which allows a decomposition of the latent space using independent subspace analysis (ISA) and demonstrate that this prior leads to disentangled representations even for the unmodified ELBO objective. This removes the trade-off between disentanglement and reconstruction loss of existing approaches.

- An even higher disentanglement of the latent space can be achieved by incorporating the proposed prior distribution into the existing approaches $\beta$-VAE and $\beta$-TCVAE. Since the prior distribution already favours a disentangled representation, the new method dominates previous in terms of the trade-off between disentanglement and model quality.

## 2 BACKGROUND

We briefly discuss previous work on variational inference in deep generative models and two modifications of the learning objective that have been proposed to learn a disentangled representation. We discuss characteristic biases of variational inference and how the modifications of the learning objective actually accentuate these biases.

### 2.1 DISENTANGLED REPRESENTATION LEARNING

**Variational Autoencoder** Kingma & Welling (2014) introduce a latent variable model that combines a generative model, the decoder, with an inference network, the encoder. Training is performed by optimizing the *evidence lower bound* (ELBO) averaged over the empirical distribution:

$$\mathcal{L}_{\text{ELBO}} = \mathbb{E}_{q_\phi(\boldsymbol{z}|\boldsymbol{x})}\left[\log p_\theta(\boldsymbol{x}|\boldsymbol{z})\right] - D_{\text{KL}}(q_\phi(\boldsymbol{z}|\boldsymbol{x})\|p(\boldsymbol{z})), \tag{1}$$

where the decoder $p_\theta(\boldsymbol{x}|\boldsymbol{z})$ is a deep learning model with parameters $\theta$ and each $\boldsymbol{z}^l$ is sampled from the encoder $\boldsymbol{z}^l \sim q_\phi(\boldsymbol{z}|\boldsymbol{x})$ with variational parameters $\phi$. When choosing appropriate families of distributions, gradients through the samples $\boldsymbol{z}^l$ can be estimated using the *reparameterization trick*. The approximate posterior $q_\phi(\boldsymbol{z}|\boldsymbol{x})$ is usually modelled as a multivariate Gaussian with diagonal covariance matrix and the prior $p(\boldsymbol{z})$ is typically the standard normal distribution.

**$\beta$-VAE** Higgins et al. (2016) propose to modify the evidence lower bound objective and penalize the KL-divergence of the ELBO:

$$\mathcal{L}_{\beta\text{-ELBO}} = \mathbb{E}_{q_\phi(\boldsymbol{z}|\boldsymbol{x})}\left[\log p_\theta(\boldsymbol{x}|\boldsymbol{z})\right] - \beta D_{\text{KL}}(q_\phi(\boldsymbol{z}|\boldsymbol{x})\|p(\boldsymbol{z})), \tag{2}$$

where $\beta > 1$ is a free parameter that should encourage a disentangled representation. In Burgess et al. (2017) the authors provide further thoughts on the mechanism that leads to these disentangled representations. However we will show in the following that this parameter introduces a trade-off between reconstruction loss and disentanglement. Furthermore, we show in section 2.4 that this parameter amplifies biases of variational inference towards orthogonalization and pruning.

$\beta$**-TCVAE** In Chen et al. (2018) the authors propose an alternative decomposition of the ELBO, that leads to the recent variant of $\beta$-VAE called $\beta$-TCVAE. They demonstrate that $\beta$-TCVAE allows to learn representations with higher MIG score than $\beta$-VAE (Higgins et al., 2016), InfoGAN (Chen et al., 2016) and FactorVAE (Kim & Mnih, 2018). The authors propose to decompose the KL-term in the ELBO objective into three parts and to weight them independently:

$$\mathbb{E}_{p_\theta(\boldsymbol{x})}\left[D_{\mathrm{KL}}(q_\phi(\boldsymbol{z}|\boldsymbol{x})\|p(\boldsymbol{z}))\right] =$$
$$= D_{\mathrm{KL}}(q_\phi(\boldsymbol{z}|\boldsymbol{x})\|q_\phi(\boldsymbol{z})p_\theta(\boldsymbol{x})) + D_{\mathrm{KL}}(q_\phi(\boldsymbol{z})\|\prod_j q_\phi(\boldsymbol{z}_j)) + \sum_j D_{\mathrm{KL}}(q_\phi(\boldsymbol{z}_j)\|p(\boldsymbol{z}_j)) . \quad (3)$$

The first term is the index-code mutual information, the second term is the total correlation and the third term the dimension-wise KL-divergence. Because the index-code mutual information can be viewed as an estimator for the mutual information between $p_\theta(\boldsymbol{x})$ and $q_\phi(\boldsymbol{z})$, the authors propose to exclude this term when reweighting the KL-term with the $\beta$ weight. In addition to the improved objective, the authors propose a quantitative evaluation score for disentanglement, the mutual information gap (MIG). They propose to first estimate the mutual information between a latent factor and an underlying generative factor of the dataset. The mutual information gap is then defined as the difference of the mutual information between the highest and second highest correlated underlying factor.

## 2.2 RELATED WORK

Recent work has shown an increased interest into learning of interpretable representations. In addition to the work mentioned already, we briefly review some of the influential papers: Chen et al. (2016) present a variant of a GAN that encourages an interpretable latent representation by maximizing the mutual information between the observation and a small subset of latent variables. The approach relies on optimizing a lower bound of the intractable mutual information. Kim & Mnih (2018) propose a learning objective equivalent to $\beta$-TCVAE, and train it with the density ratio trick (Sugiyama et al., 2012). Kumar et al. (2017) introduce a regularizer of the KL-divergence between the approximate posterior and the prior distribution. A parallel line of research proposes not to train a perfect generative model but instead to find a simpler representation of the data (Vedantam et al., 2017; Hinton et al., 2011b). A similar strategy is followed in semi-supervised approaches that require implicit or explicit knowledge about the true underlying factors of the data (Kulkarni et al., 2015; Kingma et al., 2014; Reed et al., 2014; Baydin et al., 2017; Hinton et al., 2011a; Zhu et al., 2017; Goroshin et al., 2015; Hsu et al., 2017; Denton et al., 2017).

## 2.3 ORTHOGONALIZATION AND PRUNING IN VARIATIONAL INFERENCE

There have been several interpretations of the behaviour of the $\beta$-VAE (Chen et al., 2018; Burgess et al., 2017). Here we provide a complementary perspective: that it enhances well known statistical biases in VI (Turner & Sahani, 2011) to produce disentangled, but not necessarily useful, representations. The form of these biases can be understood by considering the variational objective when written as an explicit lower-bound: the log-likelihood of the parameters minus the KL divergence between the approximate posterior and the true posterior

$$\mathcal{L}_{\mathrm{ELBO}} = \log p_\theta(\boldsymbol{x}) - D_{\mathrm{KL}}(q_\phi(\boldsymbol{z}|\boldsymbol{x})\|p_\theta(\boldsymbol{z}|\boldsymbol{x})) \quad (4)$$

From this form it is clear that VI's estimates of the parameters $\theta$ will be biased away from the maximum likelihood solution (the maximizer of the first term) in a direction that reduces the KL between the approximate and true posteriors. When factorized approximating distributions are used, VI will therefore be biased towards settings of the parameters that reduce the statistical dependence between the latent variables in the posterior. For example, this will bias learned components towards orthogonal directions in the output space as this reduces explaining away (e.g. in the factor analysis model, VI breaks the degeneracy of the maximum-likelihood solution finding the orthogonal PCA directions, see appendix B.8). Moreover, these biases often cause components to be pruned out (in the sense that they have no effect on the observed variables) since then their posterior sits at the prior, which is typically factorized (e.g. in an over-complete factor analysis model VI prunes out components to return a complete model, see appendix B.8). For simple linear models these effects are not pathological: indeed VI is arguably selecting from amongst the degenerate maximum likelihood solutions in a sensible way. However, for more complex models the biases are more

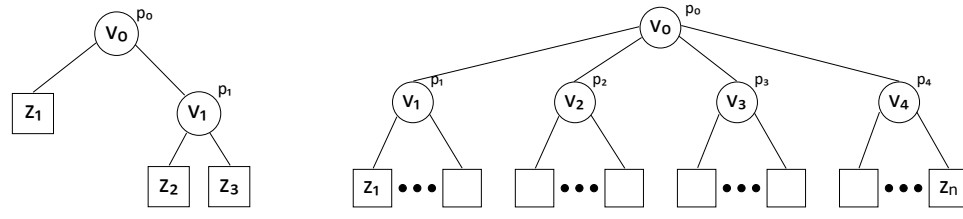

(a) Tree corresponding to Eq. 6       (b) Tree visualization of Eq. 9, an $L^p$-nested ISA model.

Figure 1: Trees representation of $L^p$-nested distributions. a) Tree of the example provided in Eq. 6. b) Tree corresponding to an $L^p$-nested ISA model.

severe: often the true posterior of the underlying model has significant dependencies (e.g. due to explaining away) and the biases can prevent the discovery of some components. For example, VAEs are known to over-prune (Burda et al., 2015; Cremer et al., 2018).

### 2.4 $\beta$-VAE EMPHASIZES ORTHOGONALIZATION AND PRUNING

What happens to these biases in the $\beta$-VAE generalization when $\beta > 1$? The short answer is that they grow. This can be understood by considering coordinate ascent of the modified objective. With $\theta$ fixed, optimising $q$ finds a solution that is closer to the prior distribution than VI due to the upweighting of the KL term in 2. With $q$ fixed, optimization over $\theta$ returns the same solution as VI (since the prior does not depend on the parameters $\theta$ and so the value of $\beta$ is irrelevant). However, since $q$ is now closer to the prior than before, the KL bias in equation 2 will be greater. These effects are shown in the ICA example in appendix B.8. VI ($\beta = 1$) learns components that are more orthogonal than the underlying ones, but $\beta = 5$ prunes out one component entirely and sets the other two to be orthogonal. This is disentangled, but arguably leads to incorrect interpretation of the data. This happens even though both methods are initialised at the true model. Arguably, the $\beta$-VAE is enhancing a statistical bug in VI and leveraging this as a feature. We believe that this can be dangerous, preventing the discovery of the underlying model.

## 3 LATENT PRIOR DISTRIBUTIONS FOR UNSUPERVISED FACTORIZATION

In this section we describe an approach for unsupervised learning of disentangled representations. Instead of modifying the ELBO-objective, we propose to use certain families of prior distributions $p(\boldsymbol{z})$, that lead to identifiable and interpretable models. In contrast to the standard normal distribution, the proposed priors are not rotationally invariant, and therefore allow interpretability of the latent space.

### 3.1 INDEPENDENT COMPONENT ANALYSIS

Independent Component Analysis (ICA) seeks to factorize a distribution into non-Gaussian factors. In order avoid the ambiguities of latent space rotations, a non-Gaussian distribution (e.g. Laplace or Student-t distribution) is used as prior for the latent variables.

**Generalized Gaussian Distribution** A generalized version of ICA (Lee & Lewicki, 2000; Zhang et al., 2004; Lewicki, 2002; Sinz & Bethge, 2010) uses a prior from the family of *exponential power distributions* of the form

$$p_{\text{ICA}}(\boldsymbol{z}) \propto \exp\left(-\tau||\boldsymbol{z}||_p^p\right) \tag{5}$$

also called *generalized Gaussian*, *generalized Laplacian* or *p-generalized normal* distribution. Using $p = 2/(1 + \kappa)$ the parameter $\kappa$ is a measure of kurtosis (Box & Tiao, 1973). This family of distributions generalizes the normal ($\kappa = 0$) and the Laplacian ($\kappa = 1$) distribution. In general we get for $\kappa > 0$ *leptokurtic* and for $\kappa < 0$ *platykurtic* distributions. The choice of a leptokurtic or platykurtic distribution has a strong influence on how a generative factor of the data is represented by a latent dimension. Fig. 2 depicts two possible prior distributions over latents that represent the

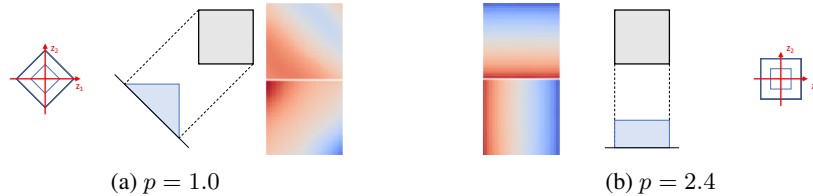

(a) $p = 1.0$        (b) $p = 2.4$

Figure 2: Leptokurtic and platykurtic priors encourage different orientations of the encoding of the (x,y) location of a sprite in the dSprites dataset. A leptokurtic distribution (here the Laplace distribution) has, in two dimensions, contour lines along diagonal directions and expects most of the probability mass around $0$. Because the (x,y) locations in dSprites are distributed in a square, the projection of the coordinates onto the diagonal fits better to the Laplace prior. A platykurtic distribution however is more similar to a uniform distribution, with axis aligned contour lines in two dimensions. This fits better to an orthogonal projection of the (x,y) location. The red and blue colour coding denotes the value of the latent variable for the respective (x,y) location of a sprite.

(x,y) spatial location of a sprite in the dSprites dataset (Matthey et al., 2017). The leptokurtic distribution expects most of the probability mass around $0$ and therefore favours a projection of the x and y coordinates, which are distributed in a square, onto the diagonal. The platykurtic prior is closer to a uniform distribution and therefore encourages an axis-aligned representation. This example shows how the choice of the prior will effect the latent representation.

Obviously the normal distribution is a special instance of the class of $L^p$-spherically symmetric distributions, and the normal distribution is the only $L^2$-spherically symmetric distribution with independent marginals. Equivalently (Sinz et al., 2009a) showed that this also generalizes to arbitrary values of $p$. The marginals of the $p$-generalized normal distribution are independent, and it is the only factorial model in the class of $L^p$-spherically symmetric distributions.

We investigate the behaviour of $L^p$-spherically symmetric distributions as prior distributions for $p(\boldsymbol{z})$ in the experiments in section 4.

## 3.2 Independent Subspace Analysis

ICA can be further generalized to include independence between subspaces, but dependencies within them, by using a more general prior, the family of $L^p$-nested symmetric distributions (Hyvärinen & Hoyer, 2000; Hyvärinen & Köster, 2007; Sinz et al., 2009b; Sinz & Bethge, 2010).

**$L^p$-nested Function** To start, let's consider functions of the form

$$\left(|z_1|^{p_0} + (|z_2|^{p_1} + |z_3|^{p_1})^{\frac{p_0}{p_1}}\right)^{\frac{1}{p_0}} , \tag{6}$$

with $p_0, p_1 \in \mathbb{R}$. This function is a cascade of two $L^p$-norms. To aid intuition we provide a visualization of this distribution in figure 1a, which depicts (6) as a tree that visualizes the nested structure of the norms. We call the class of functions which employ this structure $L^p$-*nested*.

**$L^p$-nested Distribution** Given an $L^p$-nested function $f$ and a radial density $\psi_0 : \mathbb{R} \mapsto \mathbb{R}^+$ we define the $L^p$-*nested symmetric distribution* following Fernandez et al. (1995) as

$$p_{\text{ISA}}(\boldsymbol{z}) = \frac{\psi_0(f(\boldsymbol{z}))}{f(\boldsymbol{z})^{n-1} \mathcal{S}_f(1)} , \tag{7}$$

where $\mathcal{S}_f(1)$ is the surface area of the $L^p$-nested sphere. This surface area can be obtained by using the gamma function:

$$\mathcal{S}_f(R) = R^{n-1} 2^n \prod_{i \in I} \frac{\prod_{k=1}^{l_1} \Gamma\left[\frac{n_{i,k}}{p_i}\right]}{p_i^{l_i-1} \Gamma\left[\frac{n_i}{p_i}\right]} , \tag{8}$$

where $l_i$ is the number of children of a node $i$, $n_i$ is the number of leaves in a subtree under the node $i$, and $n_{i,k}$ is the number of leaves in the subtree of the $k$-th children of node $i$. For further details we refer the reader to the excellent work of Sinz & Bethge (2010).

**Independent Subspace Analysis** The family of $L^p$-nested distributions allows a generalization of ICA called independent subspace analysis (ISA). ISA uses a subclass of $L^p$-nested distributions, which are defined by functions of the form

$$f(\boldsymbol{z}) = \left( \left( \sum_{j=1}^{n_1} |z_j|^{p_1} \right)^{\frac{p_0}{p_1}} + \cdots + \left( \sum_{j=n_1+\cdots+n_{l-1}+1}^{n} |z_j|^{p_l} \right)^{\frac{p_0}{p_l}} \right)^{\frac{1}{p_0}}, \tag{9}$$

and correspond to trees of depth two. The tree structure of this subclass of functions is visualized in figure 1b where each $v_i$, $i = 1, \ldots, l_0$ denotes the function value of the $L^p$-norm evaluated over a node's children. The components $z_j$ of $z$ that contribute to each $v_i$ form a subspace

$$\mathcal{V}_i = \left\{ z_j \,\middle|\, j = a \ldots b \text{ with } a = \sum_{k=1}^{i-1} n_k + 1,\, b = a + n_i \right\}. \tag{10}$$

Sinz & Bethge (2010) showed that the subspaces $\mathcal{V}_1, \ldots, \mathcal{V}_{l_0}$ become independent when using the radial distribution

$$\psi_0(v_0) = \frac{p_0 v_0^{n-1}}{\Gamma\left[\frac{n}{p_0}\right] s^{\frac{n}{p_0}}} \exp\left( -\frac{v_0^{p_0}}{s} \right) \tag{11}$$

which we can interpret as a generalization of the Chi-distribution.

**ISA-VAE** We propose to choose the latent prior $p_{\text{ISA}}(\boldsymbol{z})$ (Eq. 7) with $f(\boldsymbol{z})$ from the family of ISA models of the form of Eq. 9, which allows us to define independent subspaces in the latent space. The Kulback-Leibler divergence of the ELBO-objective can be estimated by Monte-Carlo sampling. This leads to an ELBO-objective of the form

$$\mathcal{L}_{\text{ISA-VAE}} = \mathbb{E}_{z \sim q_\phi(\boldsymbol{z}|\boldsymbol{x})} \left[ \log p_\theta(\boldsymbol{x}|\boldsymbol{z}) + \log p_{\text{ISA}}(\boldsymbol{z}) - \log q_\phi(\boldsymbol{z}|\boldsymbol{x}) \right], \tag{12}$$

which only requires to compute the log-density of the prior that is readily accessible from the density defined in Eq. 7. As discussed in Roeder et al. (2017) this form of the ELBO even has potential advantages (variance reduction) in comparison to a closed form KL-divergence.

**ISA-TCVAE** The proposed latent prior can also be combined with the $\beta$-TCVAE approach and we get the objective

$$\mathcal{L}_{\text{ISA-TCVAE}} = \mathbb{E}_{z \sim q_\phi(\boldsymbol{z}|\boldsymbol{x})} \left[ \log p_\theta(\boldsymbol{x}|\boldsymbol{z}) \right] - I_q(\boldsymbol{z}; \boldsymbol{x}) - \beta D_{\text{KL}}(q_\phi(\boldsymbol{z}) \| \prod_j q_\phi(\boldsymbol{z}_j)) - \sum_j D_{\text{KL}}(q_\phi(\boldsymbol{z}_j) \| p_{\text{ISA}}(\boldsymbol{z}_j)), \tag{13}$$

where $I_q$ denotes the index code mutual information. To compute the terms in Eq. 13, Chen et al. (2018) propose a Monte-Carlo sampling approach called minibatch-weighted sampling, which also only requires to compute the log density of the prior.

**Sampling and the Reparameterization Trick** If we want to sample from the generative model we have to be able to sample from the prior distribution. Sinz & Bethge (2010) describe an exact sampling approach to sample from an $L^p$-nested distribution, which we reproduce as Algorithm 1 in the appendix. Note that during training we only have to sample from the approximate posterior $q_\phi$, which we do not have to modify and which can remain a multivariate Gaussian distribution following the original VAE approach. As a consequence, the reparameterization trick can be applied (Kingma & Welling, 2014).

Experiments in the following section demonstrate that the proposed prior supports unsupervised learning of disentangled representation even for the unmodified ELBO objective ($\beta = 1$).

## 4  EXPERIMENTS

In our experiments, we evaluate the influence of the proposed prior distribution on disentanglement and on the quality of the reconstruction on the dSprites dataset (Matthey et al., 2017), which contains images of three different shapes undergoing transformations of their position, scale and rotation.

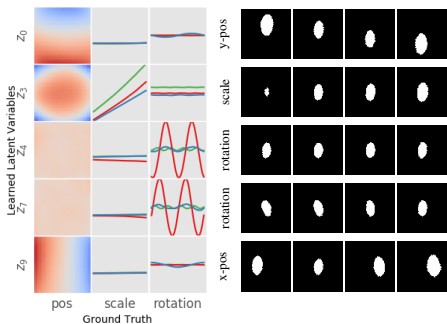 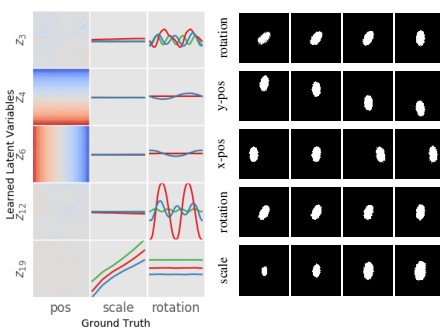

(a) ISA-VAE, $\beta = 1$, MIG: 0.20

(b) ISA-TCVAE, $\beta = 2.5$, MIG: 0.54

Figure 3: Disentangled representations for ISA-VAE and ISA-TCVAE on the dSprites dataset. We follow standard practice established in Chen et al. (2018) for visualizing latent representations and additionally show images generated by traversals of the latent along the respective axis. The red and blue colour coding in the first column denotes the value of the latent variable for the respective x,y-coordinate of the sprite in the image. Coloured lines indicate the object shape with red for ellipse, green for square, and blue for heart. (a) Even without a modification of the ELBO ($\beta = 1.0$) the proposed ISA prior leads to a disentangled representation. (b) When combining the ISA-model with $\beta$-TCVAE, a model with a high disentanglement score of MIG $= 0.54$ can be reached. This is the highest score reported for dSprites in the literature. ISA-layouts: (a) $l_0 = 5$, $l_{1,\dots,5} = 5$, $p_0 = 2.1$, $p_{1,\dots,5} = 2.2$ (b) $l_0 = 5$, $l_{1,\dots,5} = 4$, $p_0 = 2.1$, $p_{1,\dots,5} = 2.2$

**Disentanglement Metrics** To provide a quantitative evaluation of the disentanglement we compute the disentanglement metric *Mutual Information Gap* (MIG) that was proposed in Chen et al. (2018). The MIG score measures how much mutual information a latent dimension shares with the underlying factor, and how well this latent dimension is separated from the other latent factors. Therefore the MIG measures the two desired properties usually referred to with the term *disentanglement*: a factorized latent representation, and interpretability of the latent factors. Chen et al. (2018) compare the MIG metric to existing disentanglement metrics (Higgins et al., 2016; Kim & Mnih, 2018) and demonstrate that the MIG is more effective and that the other metrics do not allow to capture both properties in a desirable way.

**Reconstruction Quality** To quantify the reconstruction quality, we report the expected (log-)likelihood of the reconstructed data $\mathbb{E}_{q_\phi(z|x)} [\log p_\theta(x|z)]$. In our opinion this measure is more informative than the ELBO, frequently reported in existing work, e.g. (Chen et al., 2018), especially when varying the $\beta$ parameter, the weighting of the KL term, which is part of the ELBO.

**Comparison Baselines** Chen et al. (2018) demonstrate that $\beta$-TCVAE, a modification of the $\beta$-VAE, enables learning of representations with higher MIG score than $\beta$-VAE (Higgins et al., 2016), InfoGAN (Chen et al., 2016), and FactorVAE (Kim & Mnih, 2018). Therefore we choose to compare against $\beta$-TCVAE and $\beta$-VAE in our experiments.

**Architecture of the Encoder and Decoder** To allow an accurate comparison we use the same architecture for the decoder and encoder as presented in Chen et al. (2018). We reproduce the description of the encoder and decoder in appendix A.5

**Choosing the ISA-layout** We follow the practice of Sinz et al. (2009b) and perform a search over different values for the parameters of the ISA model and choose the model with the highest disentanglement (MIG) score. A comparison of different parameter settings and more details about this procedure are provided in appendix A.1

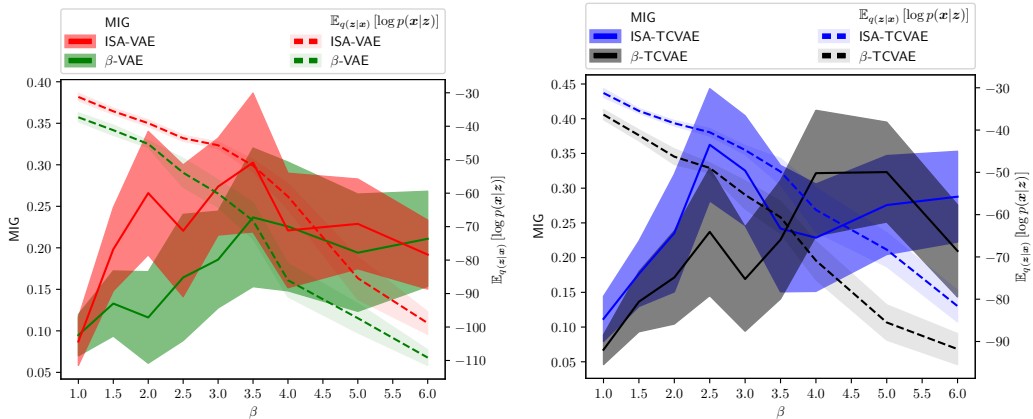

Figure 4: Comparison of the proposed ISA-VAE model with $\beta$-VAE and $\beta$-TCVAE. Evaluation is performed with respect to the quality of the disentanglement (MIG score, solid line) and reconstruction quality (dashed line) for different values of $\beta$. The proposed models ISA-VAE and ISA-TCVAE reach a disentangled representation for small values of $\beta$ which allows better reconstructions due to the trade-off between the $\beta$-weight of the KL-term and the reconstruction loss in the modified ELBO. Shaded regions depict 90% confidence intervals. Evaluated on the dSprites dataset with $n = 16$ for each value of $\beta$. ISA layout: $l_0 = 5$, $l_{1,\dots,5} = 4$, $p_0 = 2.1$, $p_{1,\dots,5} = 2.2$.

### 4.1 SUPPORT OF THE PRIOR TO LEARN DISENTANGLED REPRESENTATIONS

First, we investigate the ability of the prior to support unsupervised learning of disentangled representations for the unmodified ELBO-objective. Figure 3a depicts the structure of the latent representation after training for ISA-VAE, a combination of the $L^p$-nested ISA prior with the standard VAE approach. Because our prior allows independent subspaces the latent space becomes interpretable even when using the unmodified ELBO objective with $\beta = 1$. The plots were produced with the reference implementation of Chen et al. (2018) for visualizing latent representations for the dSprites dataset. When combining the ISA-model with $\beta$-TCVAE and varying the $\beta$ parameter, a model with a high disentanglement score of MIG $= 0.54$ can be reached. This is the so far highest score reported for dSprites in the literature.

### 4.2 QUANTITATIVE COMPARISON OF THE DIFFERENT APPROACHES

This benefit of the proposed prior, that it encourages disentangled representations becomes even more obvious in our quantitative comparison. We compare the approaches ISA-VAE and ISA-TCVAE, that use the proposed $L^p$-nested prior $p_{\text{ISA}}$ with their respective counterpart $\beta$-VAE (Higgins et al., 2016) and $\beta$-TCVAE (Chen et al., 2018) that use the standard normal prior $p_{\mathcal{N}}$. Because the amount of disentanglement depends on the choice of the parameter $\beta$, we vary $\beta$ in the interval between 1 and 6 with a stepsize of 0.5. We compare the performance of the four different approaches in figure 4 with 16 experiments for each value of $\beta$. Clearly in both cases a higher disentanglement score can be achieved already for smaller values of $\beta$ with ISA-$\beta$-VAE and ISA-$\beta$-TCVAE in comparison to the original approaches. Even when choosing the individually best value of $\beta$, that reaches the highest MIG score for each method, the poposed approaches reach a higher MIG score than their respective counterparts. Fig. 5 depicts the distribution of MIG scores for the individually best value of $\beta$: Both average and mean of the MIG scores are higher for the variants that use the ISA-model.

### 4.3 TRADE-OFF BETWEEN DISENTANGLEMENT AND RECONSTRUCTION LOSS

Since the proposed prior facilitates learning of disentangled representations, not only a higher disentanglement score can be reached, but also higher scores are reached for smaller values of $\beta$, when compared to the original approaches. This leads to a clear improvement of the trade-off between disentanglement and reconstruction loss. The improvement of this trade-off is demonstrated in figure 4, where we plot both the disentanglement score and the reconstruction loss for varying values

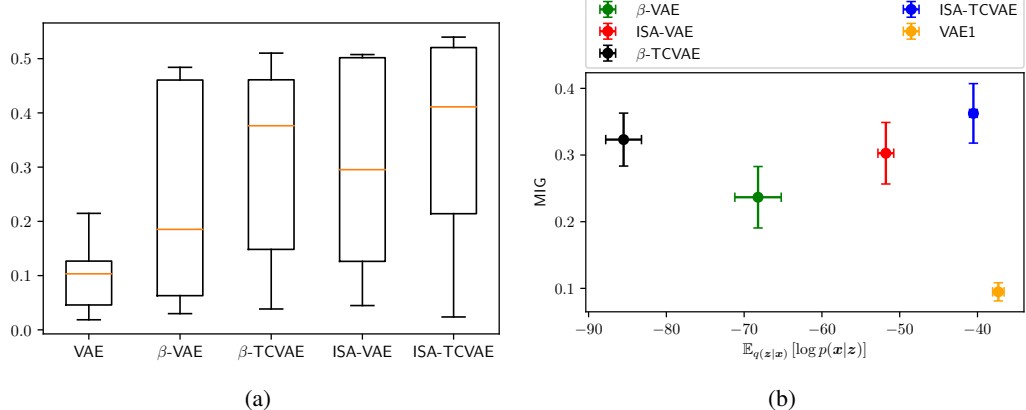

(a)                      (b)

Figure 5: Comparison of the different approaches using the optimal $\beta$ value in terms of MIG score. (a) Box plot of the distribution of MIG-scores (b) Scatter plot of MIG-score and reconstruction loss with error bars denoting the standard error. Note that when comparing $\beta$-TCVAE with $\beta$-VAE the MIG score has improved, but at the same time the reconstruction quality decreases significantly. The proposed approaches ISA-VAE and ISA-TCVAE allow a better trade-off between disentanglement and the reconstruction loss, almost reaching the reconstruction quality of the standard, non-disentangling VAE. Number of experiments: 16 each. Layout of the ISA model in all experiments: $l_0 = 5$, $l_{1,...,5} = 4$, $p_0 = 2.1$, $p_{1,...,5} = 2.2$. $\beta$-values of each approach: $\beta$-VAE: 3.5, $\beta$-TCVAE: 5, ISA-VAE: 3.5, ISA-TCVAE: 2.5

of $\beta$. ISA-$\beta$-VAE and ISA-$\beta$-TCVAE reach high values of the disentanglement score for smaller values of $\beta$ which and at the same time preserves a higher quality of the reconstruction than the respective original approaches. At the same time we observe that with the proposed prior the quality of the reconstruction decreases at a smaller rate than for the original approaches.

This improvement of the trade-off between disentanglement and the reconstruction loss becomes also obvious in figure 5b where we plot the MIG score with respect to the reconstruction loss for the individually best value of $\beta$. The proposed approaches ISA-VAE and ISA-TCVAE allow higher MIG scores than their respective base lines while at the same time providing a better reconstruction quality, almost reaching the reconstruction quality of the standard, non-disentangling VAE. Interestingly the plot also shows that the increase of the MIG score for the baseline method $\beta$-TCVAE comes at the cost of a much lower reconstruction quality. This difference in the reconstruction quality becomes obvious in the quality of the reconstructed images. Please refer to the appendix where we present latent traversals in appendix A.3 and image reconstruction experiments in appendix A.4. With the proposed approach ISA-TCVAE the reconstruction quality can be increased significantly while at the same time providing a higher disentanglement.

## 5 CONCLUSION

We presented a structured prior for unsupervised learning of disentangled representations in deep generative models. We choose the prior from the family of $L^p$-*nested symmetric distributions* which enables the definition of a hierarchy of independent subspaces in the latent space. In contrast to the standard normal prior that is often used in training of deep generative models the proposed prior is not rotationally invariant and therefore enhances the interpretability of the latent space. We demonstrate in our experiments, that a combination of the proposed prior with existing approaches for unsupervised learning of disentangled representations allows a significant improvement of the trade-off between disentanglement and reconstruction loss.

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

# A  APPENDIX

## A.1  CHOOSING THE ISA-LAYOUT

We vary $l_0$ between 4 and 10 and choose the same value for $l_1 = l_{2,\dots,l_0}$ between 2 and 10. We set the parameter range of the exponents $p_i$ to $p_i \in [0.9, 2.4]$ with a discretization step size of 0.1, which includes lepto- and platokyrtic distributions. Fig. 2 depicts how lepto- and platykurtic distributions at the child subspaces lead to different representations of the x and y coordinate. Because the MIG metric evaluates axis-alignment of the latent dimensions to the underlying factors, here the x and y coordinate, platykurtic priors in general achieve a higher MIG score. The child subspaces share the same parameter $p_1 = p_{2,\dots,l_0}$ and we choose the exponent of the root node as $p_0 \neq p_1$ to ensure independence of the subspaces. To study the influence of the layout on the reconstruction quality and MIG score we compare the results for different values of $p_0$, $p_{1,\dots,5}$ and $l_1$, and vary the value of $\beta$ in the interval $\beta \in [1, 4]$ with a step size of 0.5 and repeat each experiment four times. We compare four layouts with the highest MIG score for each subspace layout in Fig. 6 where we plot the mean and standard error of MIG score and reconstruction loss.

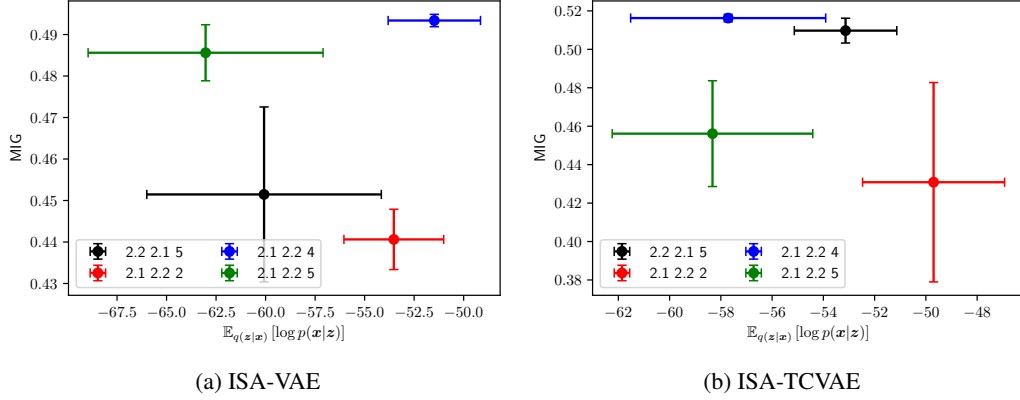

(a) ISA-VAE

(b) ISA-TCVAE

Figure 6: Disentanglement score (MIG) with respect to the reconstruction quality for different layouts of the ISA subspaces. The number of subspaces $l_0 = 5$ is constant throughout all experiments. For this dataset, the confguration $p_0 = 2.1$, $p_{1,\dots,5} = 2.2$ and $l_1 = 4$ (denoted in black) is most appropriate as it achieves high MIG scores while maintaining a good reconstruction quality, both for the ISA-VAE and the ISA-TCVAE model.

## A.2 Sampling from $L^p$-nested Symmetric Distributions

---

**Algorithm 1:** Exact sampling algorithm for $L^p$-nested symmetric distributions from Sinz & Bethge (2010)

---

**Input** : The radial distribution $\psi_0(v_0)$ of an $L^p$-nested symmetric distribution $p_{L^p}$ for the $L^p$-nested function $f$

**Output:** Sample $x$ from $p_{L^p}$

1. Sample $v_0$ from a beta distribution $\beta[n, 1]$

2. For each inner node $i$ of the tree associated with $f$, sample the auxiliary variable $s_i$ from a Dirichlet distribution $\mathrm{Dir}\left[\frac{n_{i,1}}{p_i}, \ldots, \frac{n_{i,l_1}}{p_i}\right]$ where $n_{i,k}$ are the number of leaves in the subtree under node $i, k$. Obtain coordinates on the $L^p$-nested sphere within the positive orthant by $s_i \mapsto s_i^{\frac{1}{p_i}} = \tilde{u}_i$ (the exponentiation is taken component-wise)

3. Transform these samples to Cartesian coordinates by $v_i \cdot \tilde{u}_i = v_{i,1:l_i}$ for each inner node, starting from the root node and descending to lower layers. The components of $v_{i,1:l_i}$ constitute the radii for the layer direct below them. If $i = 0$, the radius had been sampled in step 1

4. Once the two previous steps have been repeated until no inner node is left, we have a sample $x$ from the uniform distribution in the positive quadrant. Normalize $x$ to get a uniform sample from the sphere $u = \frac{x}{f(x)}$

5. Sample a new radius $\tilde{v}_0$ from the radial distribution of the target radial distribution $\psi_0$ and obtain the sample via $\tilde{x} = \tilde{v}_0 \cdot u$

6. Multiply each entry $x_i$ of $\tilde{x}$ by and independent sample $z_i$ from the uniform distribution over $\{-1, 1\}$.

---

## A.3 Disentanglement Representations and Latent Traversals

We follow standard practice established in Chen et al. (2018) for visualizing latent representations and additionally show images generated by traversals of the latent along the respective axis. The red and blue color coding in the first column denotes the value of the latent variable for the respective x,y-coordinate of the sprite in the image. Colored lines indicate the object shape with red for ellipse, green for square, and blue for heart.

ISA-layouts:
ISA-VAE: $l_0 = 5, l_{1,\ldots,5} = 5, p_0 = 2.1, p_{1,\ldots,5} = 2.2$
ISA-TCVAE $l_0 = 5, l_{1,\ldots,5} = 4, p_0 = 2.1, p_{1,\ldots,5} = 2.2$

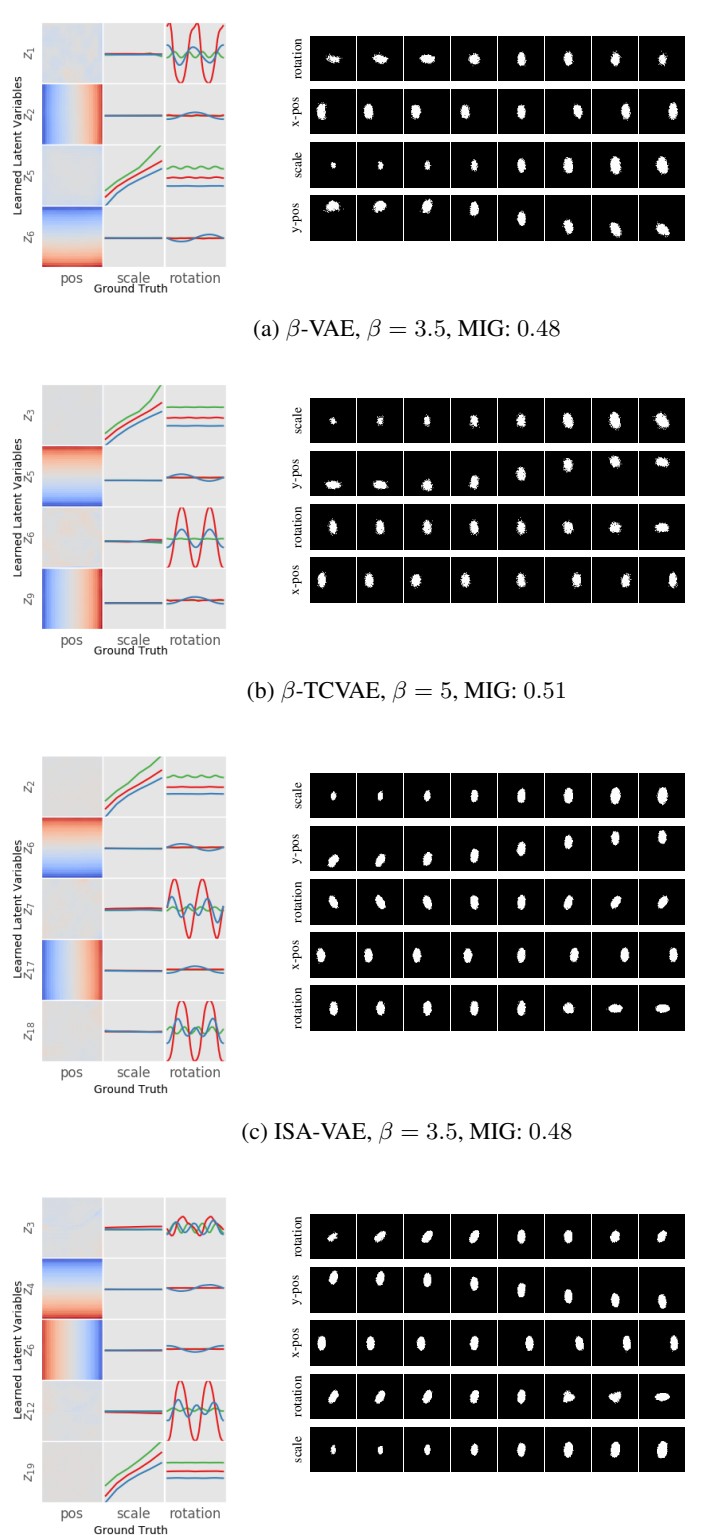

(a) $\beta$-VAE, $\beta = 3.5$, MIG: 0.48

(b) $\beta$-TCVAE, $\beta = 5$, MIG: 0.51

(c) ISA-VAE, $\beta = 3.5$, MIG: 0.48

(d) ISA-TCVAE, $\beta = 2.5$, MIG: 0.54

Figure 7: Disentangled representations for $\beta$-VAE, $\beta$-TCVAE, ISA-VAE and ISA-TCVAE and latent traversals for the ellipse shape.

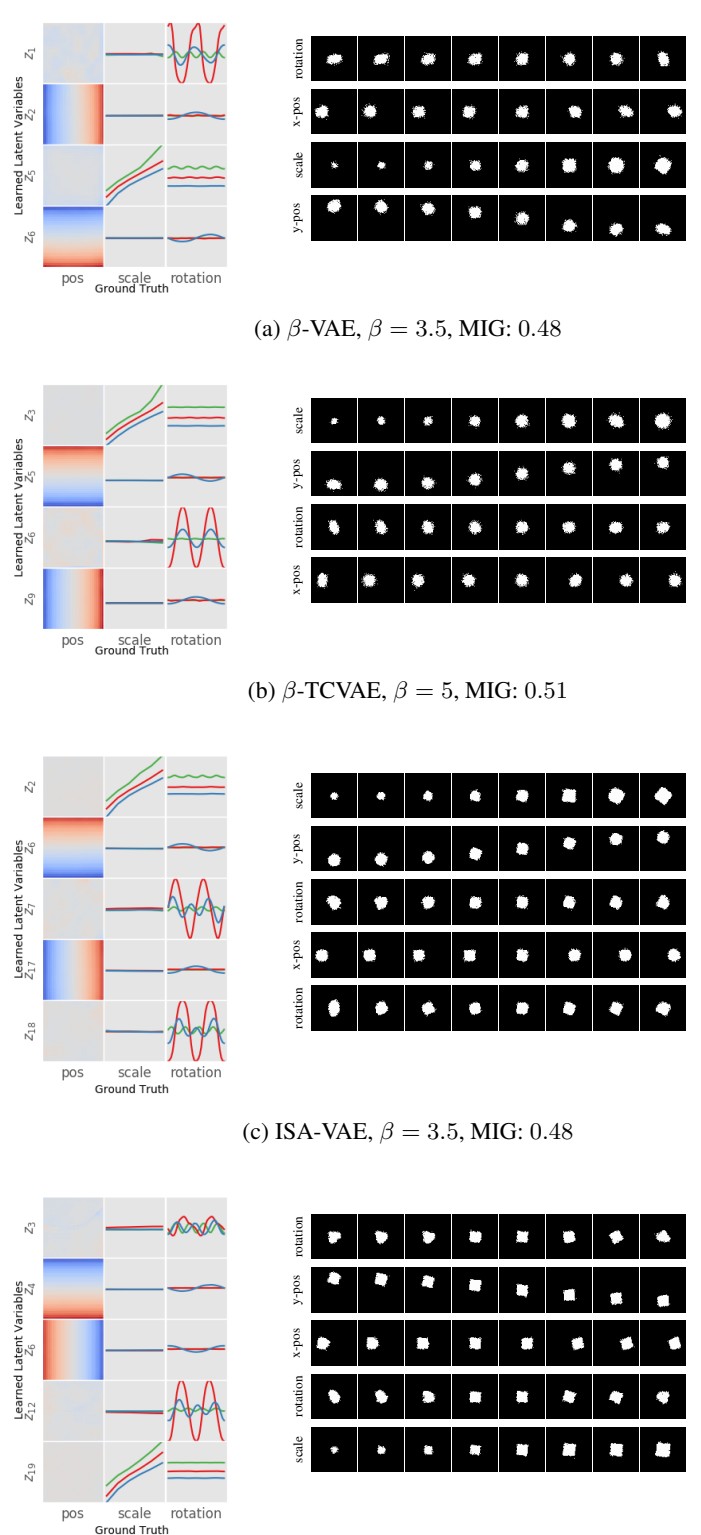

(a) $\beta$-VAE, $\beta = 3.5$, MIG: 0.48

(b) $\beta$-TCVAE, $\beta = 5$, MIG: 0.51

(c) ISA-VAE, $\beta = 3.5$, MIG: 0.48

(d) ISA-TCVAE, $\beta = 2.5$, MIG: 0.54

Figure 8: Disentangled representations for $\beta$-VAE, $\beta$-TCVAE, ISA-VAE and ISA-TCVAE and latent traversals for the square shape.

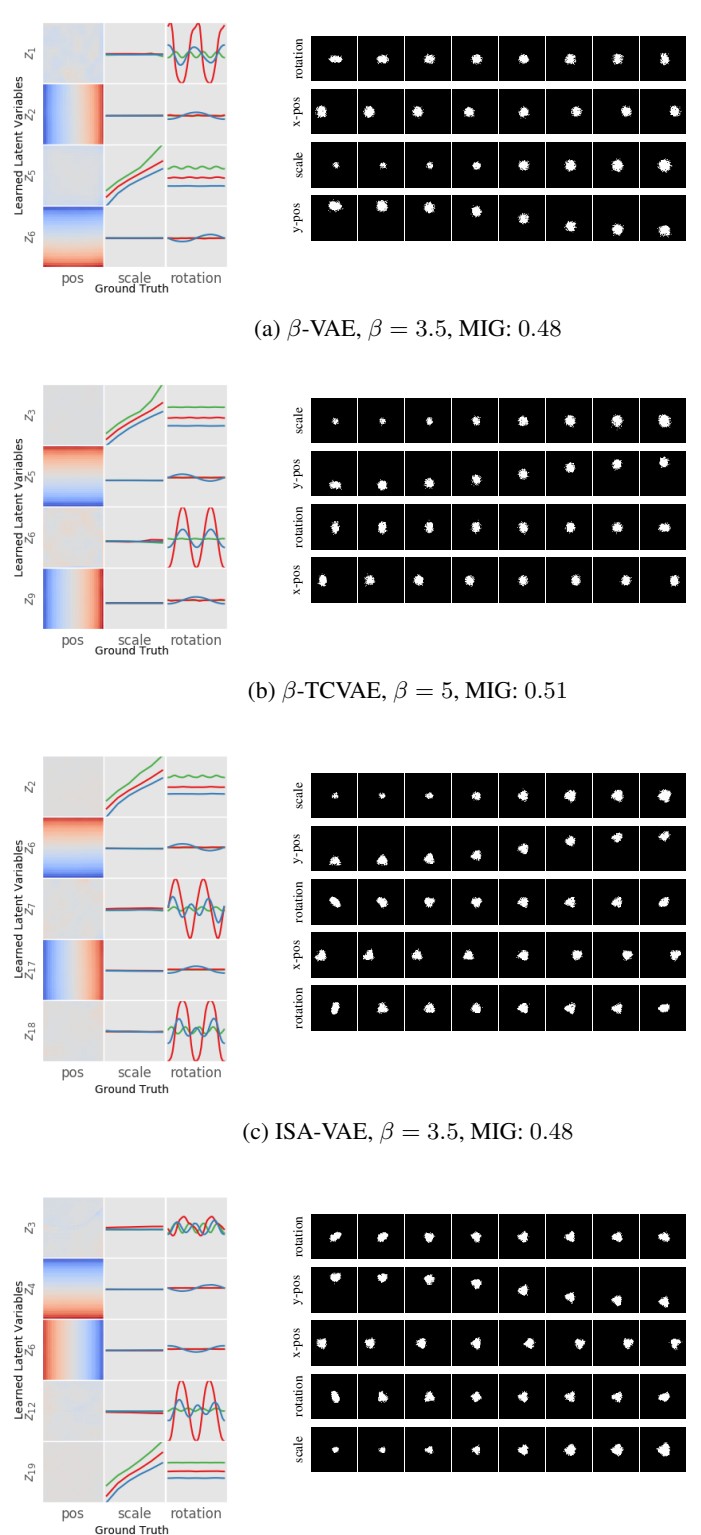

(a) $\beta$-VAE, $\beta = 3.5$, MIG: 0.48

(b) $\beta$-TCVAE, $\beta = 5$, MIG: 0.51

(c) ISA-VAE, $\beta = 3.5$, MIG: 0.48

(d) ISA-TCVAE, $\beta = 2.5$, MIG: 0.54

Figure 9: Disentangled representations for $\beta$-VAE, $\beta$-TCVAE, ISA-VAE and ISA-TCVAE and latent traversals for the heart shape.

## A.4 Image Reconstruction Results

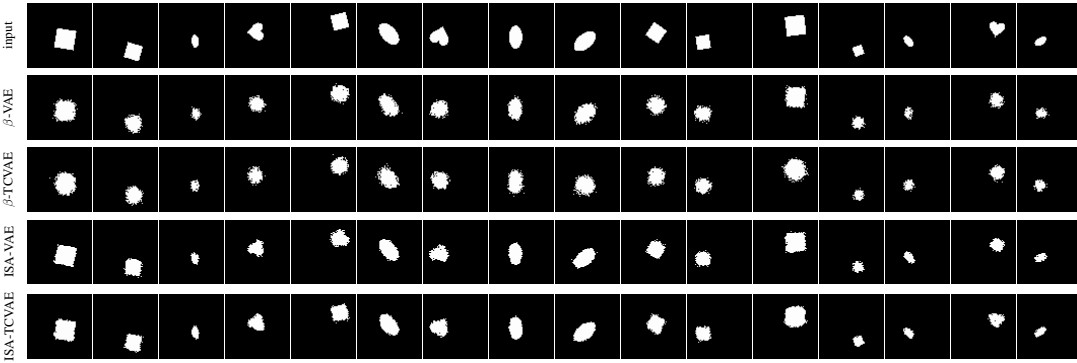

Figure 10: Reconstructed images for $\beta$-VAE, $\beta$-TCVAE, ISA-VAE and ISA-TCVAE using the models from figure 5 and appendix A.3. Note that because of the trade-off between disentanglement and reconstruction loss the images reconstructed with $\beta$-VAE and $\beta$-TCVAE appear much noisier than the models with the ISA prior. Further, the ISA prior allows to reconstruct more details of the heart shape than $\beta$-VAE and $\beta$-TCVAE.

## A.5 Model Architecture (PyTorch)

The models were trained with the optimization algorithm Adam (Kingma & Ba, 2015) using a learning rate parameter of 0.01

All unmentioned hyperparameters are PyTorch v0.41 defaults.

```python
class MLPEncoder(nn.Module):
    def __init__(self, output_dim):
        super(MLPEncoder, self).__init__()
        self.output_dim = output_dim

        self.fc1 = nn.Linear(4096, 1200)
        self.fc2 = nn.Linear(1200, 1200)
        self.fc3 = nn.Linear(1200, output_dim)

        self.conv_z = nn.Conv2d(64, output_dim, 4, 1, 0)

        # setup the non-linearity
        self.act = nn.ReLU(inplace=True)

    def forward(self, x):
        h = x.view(-1, 64 * 64)
        h = self.act(self.fc1(h))
        h = self.act(self.fc2(h))
        h = self.fc3(h)
        z = h.view(x.size(0), self.output_dim)
        return z

class MLPDecoder(nn.Module):
    def __init__(self, input_dim):
        super(MLPDecoder, self).__init__()
        self.net = nn.Sequential(
            nn.Linear(input_dim, 1200),
            nn.Tanh(),
            nn.Linear(1200, 1200),
```

```
            nn.Tanh(),
            nn.Linear(1200, 1200),
            nn.Tanh(),
            nn.Linear(1200, 4096)
        )

    def forward(self, z):
        h = z.view(z.size(0), -1)
        h = self.net(h)
        mu_img = h.view(z.size(0), 1, 64, 64)
        return mu_img
```

Architecture of the encoder and decoder which is identical to the architecture in Chen et al. (2018).

# B TOY EXAMPLES SHOWING BIASES IN VI AND $\beta$-VI

This section provides the details of the toy examples that reveal the biases in variational methods.

First we will consider the factor analysis model showing that VI breaks the degeneracy of the maximum-likelihood solution to 1) discover orthogonal weights that lie in the PCA directions, 2) prune out extra components in over-complete factor analysis models, even though there are solutions with the same likelihood that preserve all components. We also show that in these examples the $\beta$-VI returns identical model fits to VI regardless of the setting of $\beta$.

Second, we consider an over-complete ICA model and initialize using the true model. We show that 1) VI is biased away from the true component directions towards more orthogonal directions, and 2) $\beta$-VI with a modest setting of $\beta = 5$ prunes away one of the components and finds orthogonal directions for the other two. That is, it finds a disentangled representation, but one which does not reflect the underlying components.

## B.1 BACKGROUND

The $\beta$-VAE optimizes the modified free-energy, $\mathcal{F}_\beta(q(z_{1:N}), \theta)$, with respect to the parameters $\theta$ and the variational approximation $q(z_{1:N})$,

$$\mathcal{F}_\beta(q(z_{1:N}), \theta) = \mathbb{E}_{q(z_{1:N})}(\log p(x_{1:N}|z_{1:N}, \theta)) - \beta \mathrm{KL}(q(z_{1:N})||p(z_{1:N})).$$

Consider the case where $M = \frac{1}{\beta}$ is a positive integer, $M \in \mathbb{N}$, we then have

$$\mathcal{F}_\beta(q(z_{1:N}), \theta) = \sum_{n=1}^N \left[ \mathbb{E}_{q(z_n)}(M(\beta) \log p(x_n|z_n, \theta)) - \mathrm{KL}(q(z_n)||p(z_n)) \right]$$

In this case, the $\beta$-VAE can be thought of as attaching $M$ replicated observations to each latent variable $z_n$ and then running standard variational inference on the new replicated dataset. This can equivalently be thought of as raising each likelihood $p(x_n|z_n, \theta)$ to the power $M$.

Now in real applications $\beta$ will be set to a value that is greater than one. In this case, the effect of $\beta$ is the opposite: it is to reduce the number of effective data points per latent variable to be less than one $M < 1$. Or equivalently we raise each likelihood term to a power $M$ that is less than one. Standard VI is then run on these modified data (e.g. via joint optimization of $q$ and $\theta$).

Although this view is mathematically straightforward, the perspective of the $\beta$-VAE i) modifying the dataset, and ii) applying standard VI, is useful as it will allow us to derive optimal solutions for the variational distribution $q(z)$ in simple cases like the factor analysis model considered next.

## B.2 FACTOR ANALYSIS

Consider the factor analysis generative model. Let $\mathbf{x} \in \mathbb{R}^L$ and $\mathbf{z} \in \mathbb{R}^K$.

$$\begin{aligned} &\text{for } n = 1...N \\ &\mathbf{z}_n \sim \mathcal{N}(\mathbf{0}, \mathrm{I}), \\ &\mathbf{x}_n \sim \mathcal{N}(W\mathbf{z}_n, D) \text{ where } D = \mathrm{diag}([\sigma_1^2, ..., \sigma_\mathrm{D}^2]) \end{aligned} \tag{14}$$

The true posterior is a Gaussian $p(\mathbf{z}_n|\mathbf{x}_n, \theta) = \mathcal{N}(\mathbf{z}; \mu_{\mathbf{z}|\mathbf{x}}, \Sigma_{\mathbf{z}|\mathbf{x}})$ where

$$\mu_{\mathbf{z}|\mathbf{x}} = \Sigma_{\mathbf{z}|\mathbf{x}} W^\top D^{-1} \mathbf{x} \text{ and } \Sigma_{\mathbf{z}|\mathbf{x}} = (W^\top D^{-1} W + \mathrm{I})^{-1}. \tag{15}$$

The true log-likelihood of the parameters is

$$\begin{aligned} \log p(\mathbf{x}_{1:N}|\theta) &= \sum_{n=1}^N \log \mathcal{N}(\mathbf{x}_n, \mathbf{0}, WW^\top + D) \\ &= -\frac{N}{2} \log \det(2\pi(WW^\top + D)) - \frac{1}{2} \sum_{n=1}^N \mathbf{x}_n^\top (WW^\top + D)^{-1} \mathbf{x}_n \\ &= -\frac{1}{2} N \left[ \log \det(2\pi(WW^\top + D)) + \mathrm{trace}((WW^\top + D)^{-1}(\mu_{\mathbf{x}}\mu_{\mathbf{x}}^\top + \Sigma_{\mathbf{x}})) \right] \end{aligned}$$

Here we have defined the empirical mean and covariance of the observations $\mu_{\mathbf{x}} = \frac{1}{N} \sum_{n=1}^{N} \mathbf{x}_n$ and $\Sigma_{\mathbf{x}} = \frac{1}{N} \sum_{n=1}^{N} (\mathbf{x}_n - \mu_{\mathbf{x}})(\mathbf{x}_n - \mu_{\mathbf{x}})^\top$ i.e. the sufficient statistics.

The true likelihood is invariant under orthogonal transformations of the latent variables: $\mathbf{z}' = R\mathbf{z}$ where $RR^\top = \mathrm{I}$.

Interpreting $\beta$-VI as running VI in a modified generative model (see previous section) we have the new generative process

$$
\begin{aligned}
&\text{for } n = 1...N \\
&\quad \mathbf{z}_n \sim \mathcal{N}(\mathbf{z}_n; \mathbf{0}, \mathrm{I}), \\
&\quad \text{for } m = 1...M(\beta) \\
&\qquad \mathbf{x}_{n,m} \sim \mathcal{N}(W\mathbf{z}_n, D) \text{ where } D = \mathrm{diag}([\sigma_1^2, ..., \sigma_{\mathrm{D}}^2])
\end{aligned}
$$

We now observe data and set $\mathbf{x}_{n,m} = \mathbf{x}_n$.

The posterior is again Gaussian $p(\mathbf{z}_n|\mathbf{x}_n, \theta, M(\beta)) = \mathcal{N}(\mathbf{z}_n; \tilde{\mu}_{\mathbf{z}|\mathbf{x}}(\beta, n), \tilde{\Sigma}_{\mathbf{z}|\mathbf{x}}(\beta))$ where

$$
\tilde{\mu}_{\mathbf{z}|\mathbf{x}}(\beta, n) = \tilde{\Sigma}_{\mathbf{z}|\mathbf{x}}^{-1}(\beta) M(\beta) W^\top D^{-1} \mathbf{x}_n \text{ and } \tilde{\Sigma}_{\mathbf{z}|\mathbf{x}}(\beta) = (M(\beta) W^\top D^{-1} W + \mathrm{I})^{-1}
$$

Here we have taken care to explicitly reveal all of the direct dependencies on $\beta$.

Mean-field variational inference, $q(\mathbf{z}_n) = \prod_k q_{n,k}(z_{k,d})$, will return a diagonal Gaussian approximation to the true posterior with the same mean and matching diagonal precision,

$$
q(\mathbf{z}_n|\mathbf{x}_n, \theta, M(\beta)) = \mathcal{N}\left(\mathbf{z}_n; \tilde{\mu}_{\mathbf{z}|\mathbf{x}}(\beta, n), \Sigma_q(\beta)\right) \text{ where } \Sigma_q^{-1}(\beta) = \mathrm{diag}\left(\tilde{\Sigma}_{\mathbf{z}|\mathbf{x}}^{-1}(\beta)\right)
$$

We notice that the posterior mean is a linear combination of the observations $\tilde{\mu}_{\mathbf{z}|\mathbf{x}}(\beta, n) = R(\beta)\mathbf{x}_n$ where $R(\beta) = \tilde{\Sigma}_{\mathbf{z}|\mathbf{x}}(\beta) M(\beta) W^\top D^{-1}$ are recognition weights. Notice that the recognition weights and the posterior variances are the same for all data points: they do not depend on $n$. The free-energy is then

$$
\mathcal{F}(q, \theta, \beta) = \mathbb{E}_{q(z)}(\log p(x|z)) - \mathrm{KL}(q(z)|p(z))
$$

with the reconstruction term being

$$
\begin{aligned}
\mathbb{E}_{q(z)}(\log p(x|z)) = & -\frac{1}{2\beta} \sum_{n=1}^{N} \mathbf{x}_n^\top (D^{-1} - 2R^\top W^\top D^{-1} + R^\top W^\top D^{-1} W R)\mathbf{x}_n \\
& -\frac{N}{2\beta} \log \det(2\pi D) - \frac{N}{2\beta} \mathrm{trace}(W^\top D^{-1} \Sigma_q) \\
= & -\frac{N}{2\beta} \bigg( \mathrm{trace}\left((D^{-1} - 2R^\top W^\top D^{-1} + R^\top W^\top D^{-1} W R)(\Sigma_{\mathbf{x}} + \mu_{\mathbf{x}}\mu_{\mathbf{x}}^\top)\right) \\
& + \log \det(2\pi D) + \mathrm{trace}(W^\top D^{-1} W \Sigma_q) \bigg)
\end{aligned}
\tag{16}
$$

and the KL or regularization term being

$$
\begin{aligned}
\mathrm{KL}(q(z)|p(z)) = & -\frac{NK}{2} - \frac{N}{2} \log \det(\Sigma_q) + \frac{N}{2} \mathrm{trace}(\Sigma_q) + \frac{1}{2} \sum_{n=1}^{N} \mathbf{x}_n^\top R^\top R \mathbf{x}_n \\
= & -\frac{N}{2} \left( K + \log \det(\Sigma_q) - \mathrm{trace}(\Sigma_q) - \mathrm{trace}(R^\top R(\Sigma_{\mathbf{x}} + \mu_{\mathbf{x}}\mu_{\mathbf{x}}^\top))\right).
\end{aligned}
$$

We will now consider the objective functions and the posterior distributions in several cases to reason about the parameter estimates arising from the methods above.

### B.3 EXPERIMENT 1: MEAN FIELD VI APPLIED TO FACTOR ANALYSIS YIELDS THE PCA DIRECTIONS

Consider the situation where we know a maximum likelihood solution of the weights $W_{\mathrm{ML}}$. For simplicity we select the solution $W_{\mathrm{ML}}$ which has orthogonal weights in the observation space. We

then rotate this solution by an amount $\theta$ so that $W'_{\mathrm{ML}} = R(\theta) W_{\mathrm{ML}}$. The resulting weights are no longer orthogonal (assuming the rotation is not an integer multiple of $\pi/2$). We compute the log-likelihood (which will not change) and the free-energy (which will change) and plot the true and approximate posterior covariance (which does not depend on the datapoint value $x_n$).

First here are the weights are aligned with the true ones. The log-likelihood and the free-energy take the same value of -17.82 nats.

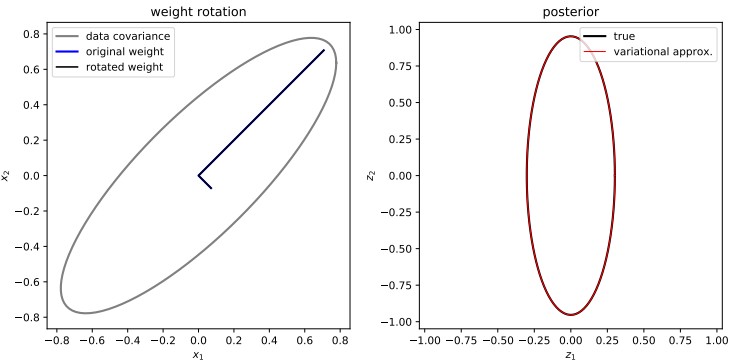

Second, here are the weights rotated $\pi/4$ and the log-likelihood is -17.82 nats and the free-energy -57.16 nats.

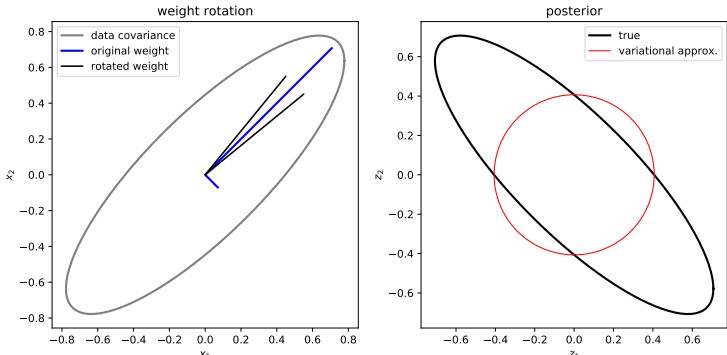

When varying the rotation away fom the orthogonal setting, $\theta$, the plots above indicate that orthogonal settings of the weights ($\theta = m\pi/2$ where $m = 0, 1, 2, ...$) lead to factorized posteriors. In these cases the KL between the approximate posterior and the true posterior is zero and the free-energy is equal to the log-likelihood. This will be the optimal free-energy for any weight setting (due to the fact that it is equal to the true log-likelihood which is maximal, and the free-energy is a lower bound of this quantity.) For intermediate values of $\theta$ the posterior is correlated and the free-energy is not tight to the log likelihood.

Now let's plot the free-energy and the log-likelihood as $\theta$ is varied. This shows that the free-energy prefers orthogonal settings of the weights as this leads to factorized posteriors, even though the log-likelihood is insensitive to $\theta$. So, variational inference recovers the same weight directions as the PCA solution.

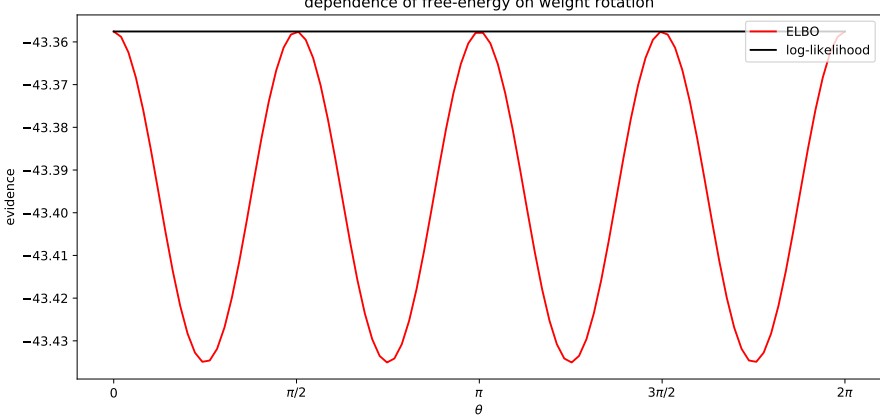

The above shows that the bias inherent in variational methods will cause them to break the symmetry in the log-likelihood and find orthogonal latent components. This occurs because orthoginal components result in posterior distributions that are factorized. These are then well-modelled by the variational approximation and result in a small KL between the approximate and true posteriors.

### B.4    EXPERIMENT 2: MEAN FIELD VI APPLIED TO OVER-COMPLETE FACTOR ANALYSIS PRUNES OUT THE ADDITIONAL LATENT DIMENSIONS

A similar effect occurs if we model 2D data with a 3D latent space. Many settings of the weights attain the maximum of the likelihood, including solutions which use all three latent variables. However, the optimal solution for VI is to retain two orthogonal components and to set the magnitude of the third component to zero. This solution a) returns weights that maximise the likelihood, and b) has a factorised posterior distribution (the pruned component having a posterior equal to its prior) that therefore incurs no cost $\mathrm{KL}(q(\mathbf{z})||p(\mathbf{z}|\mathbf{x},\theta)) = 0$. In this way the bound becomes tight.

Here's an example of this effect. We consider a model of the form:

$$\mathbf{x} = \frac{\alpha}{\sqrt{2}} \left[ \begin{array}{c} 1 \\ 1 \end{array} \right] z_1 + \frac{\beta}{\sqrt{2}} \left[ \begin{array}{c} 1 \\ 1 \end{array} \right] z_2 + \frac{\rho}{\sqrt{2}} \left[ \begin{array}{c} 1 \\ -1 \end{array} \right] z_3 + \epsilon \qquad (17)$$

We set $\alpha^2 + \beta^2 = 1$ so that all models imply the same covariance and set this to be the maximum likelihood covariance by construction. We then consider varying $\alpha$ from 0 to $1/2$. The setting equal to 0 attains the maximum of the free-energy, even though it has the same likelihood as any other setting.

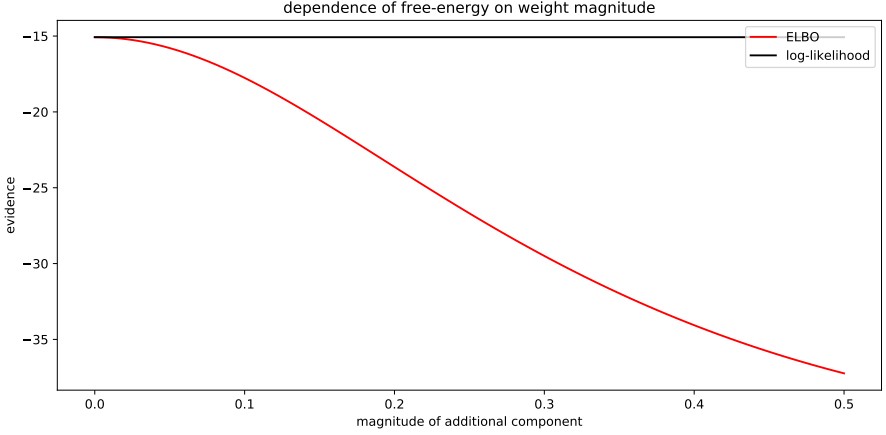

### B.5 EXPERIMENT 3: THE $\beta$-VAE ALSO YIELDS THE PCA COMPONENTS, CHANGING $\beta$ HAS NO EFFECT ON THE DIRECTION OF THE ESTIMATED COMPONENTS IN THE FA MODEL

How does the setting of $\beta$ change things? Here we rerun experiment 1 for different values of $\beta$.

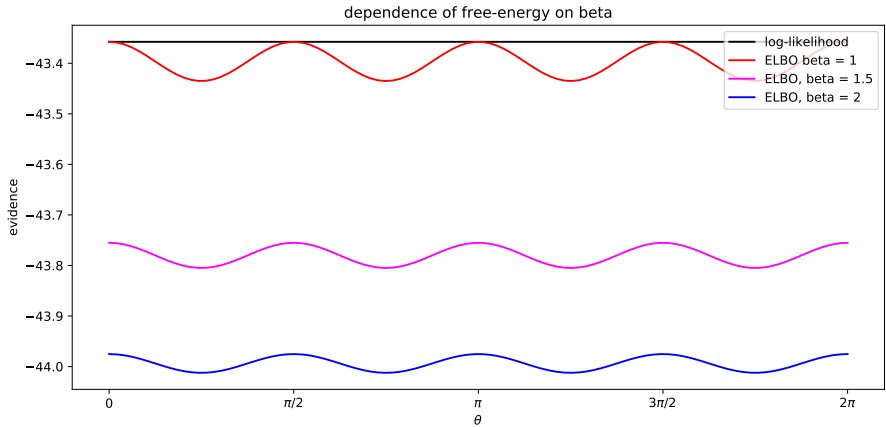

In this example, changing $\beta$ in this example just reduces the amplitude of the fluctuations in the free-energy, but it does not change the directions found. A similar observation applies to the pruning experiment.

Increasing $\beta$ will increase the uncertainty in the posterior as it is like reducing the number of observations (or increasing the observation noise, from the perspective of $q$).

### B.6 SUMMARY OF FACTOR ANALYSIS EXPERIMENTS

The behaviours introduced by the $\beta$-VAE appear relatively benign, and perhaps even helpful, in the linear case: VI is breaking the degeneracy of the maximum likelihood solution in a sensible way: selecting amongst the maximum likelihood solutions to find those that have orthogonal components and removing spurious latent dimensions. This should be tempered by the fact that the $\beta$ generalization recovered precisely the same solutions and so it was necessary to obtain the desired behaviour in the PCA case.

Similar effects will occur in deep generative models, not least since these typically also have a Gaussian prior over latent variables, and these latents are initially linearly transformed, thereby resulting in a similar degeneracy to factor analysis.

However, the behaviours above benefited from the fact that maximum-likelihood solutions could be found in which the posterior distribution over latent variables factorized. In real world examples, for example in deep generative models, this will not be case. In such cases, these same effects will cause the variational free-energy and its $\beta$-generalization to **bias the estimated parameters far away from maximum-likelihood settings, toward those settings that imply factorized Gaussian posteriors over the latent variables**.

### B.7 INDEPENDENT COMPONENT ANALYSIS

We now apply VI and the $\beta$ free-energy method to ICA. We're interested the properties of the variational objective and the $\beta$-VI objective and so we 1. fit the data using the true generative model to investigate the biases in VI and $\beta$-VI 2. do not use amortized inference, just optimizing the approximating distributions for each data point (this is possible for these small examples).

The linear independent component analysis generative model we use is defined as follows. Let $\mathbf{x} \in \mathbb{R}^L$ and $\mathbf{z} \in \mathbb{R}^K$.

$$\text{for } n = 1...N$$
$$\text{for } k = 1...K$$
$$z_{n,k} \sim \text{Student-t}(0, \sigma, v),$$
$$\mathbf{x}_n \sim \mathcal{N}(W\mathbf{z}_n, D) \text{ where } D = \text{diag}([\sigma_1^2, ..., \sigma_D^2])$$

We apply mean-field variational inference, $q(\mathbf{z}_n) = \prod_k q_{n,k}(z_{k,d})$, and use Gaussian distributions for each factor $q_{n,k}(z_{n,k}) = \mathcal{N}(z_{n,k}; \mu_{n,k}, \sigma_{n,k}^2)$.

The free-energy is computed as follows: The reconstruction term is identical to PCA: an avergage of a quadratic form wrt to a Gaussian, which is analytic. The KL is broken down into the differential entropy of q which is also analytic and the cross-entropy with the prior which we evaluate by numerical integration (finite differences). There is a cross-entropy term for each latent variable which is one reason why the code is slow (requiring N 1D numerical integrations). The gradient of the free-energy wrt the parameters $W$ and the means and variances of the Gaussian q distributions are computed using autograd.

In order to be as certain as possible that we are finding a global maximum of the free-energies, all experiments initialise at the true value of the parameters and then ensure that each gradient step improves the free-energy. Stochastic optimization or a procedure that accepted all steps regardless of the change in the objective would be faster, but they might also move us into the basis of attraction of a worse (local) optima.

### B.8 EXPERIMENT 1: LEARNING IN OVER-COMPLETE ICA

Now we define the dataset. We use a very sparse Student's t-distribtion with $v = 3.5$. For $v < 4$ the the kurtosis is undefined so the model is fairly simple to estimate model (it's a long way away from the degenerate factor analysis case which is recovered in the limit $v \to \infty$).

We use three latent components and a two dimensional observed space. The directions of the three weights are shown in blue below with data as blue circles.

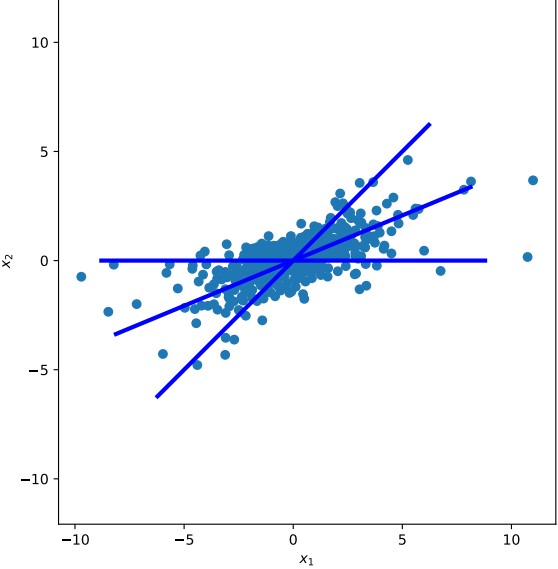

First we run variational inference finding components (shown in red below) which are more orthogonal than the true directions. This bias is in this directions as this reduces the dependencies (explaining away) in the underlying posterior.

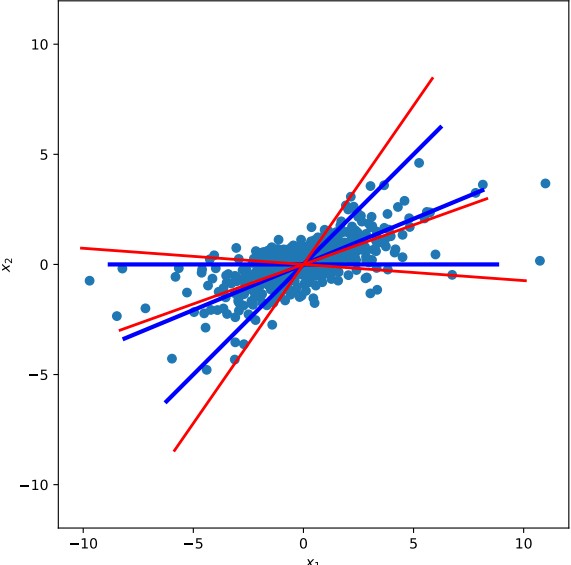

Second we run $\beta$-VI with $\beta = 5$. Two components are now found that are orthogonal with one component pruned from the solution.

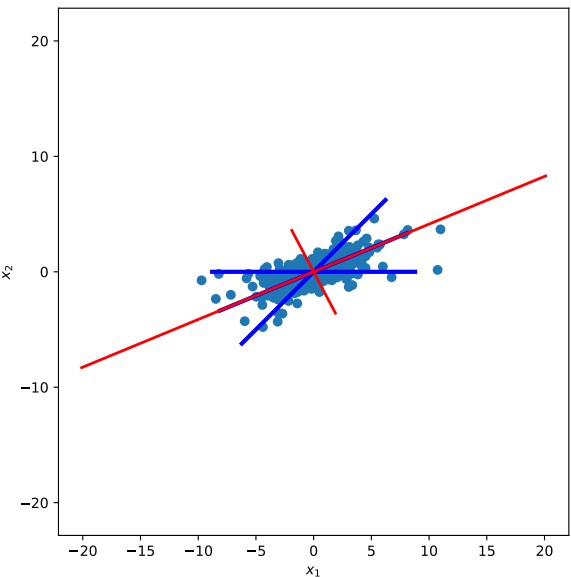

In this case the bias is so great that the true component directions are not discovered. Instead the components are forced into the orthogonal setting regardless of the structure in the data.

### B.9    SUMMARY OF INDEPENDENT COMPONENT ANALYSIS EXPERIMENT

The ICA example illustrates that this approach – of relying on a bias inherent in VI to discover meaningful components – will sometimes return meaningful structure (e.g. in the PCA experiments above). However it does not seem to be a sensible way of doing so in general. For example, explaining away often means that the true components will be entangled in the posterior, as is the case in the ICA example, and the variational bias will then move us away from this solution. The $\beta$-VI generalisation only enhances this undesirable bias.

