# OpenReview forum: "ISA-VAE: Independent Subspace Analysis with Variational Autoencoders"
_ICLR.cc/2019/Conference_

### Official Review · AnonReviewer2 · 2018-10-31
**Overall method is not presented**

**Rating:** 4
**Confidence:** 5

**Review:**

This paper presents a methodology to bring together independent subspace analysis and variational auto-encoders. Naturally, in order to do that, the authors propose a specific family of prior distributions that lead to subspace independence the Lp-nested distribution family. This prior distribution is then used to learn disentangled and interpretable representations. The mutual information gap is taken as the measure of disentanglement, while the reconstruction loss measures the quality of the representation. Experiments on the sPrites dataset are reported, and comparison with the state of the art shows some interesting results.

I understand the limitations of current approaches for learning disentangled representations, and therefore agree with the motivation of the manuscript, and in particular the choice of the prior distribution. However, I did not find the answer to some important questions, and generally speaking I believe that the contribution is not completely and clearly described.
P1) What is the shape of the posterior distribution?
P2) How does the reparametrization trick work in your case?
P3) How can one choose the layout of the subspaces, or this is also learned?

Moreover, and this is crucial, the proposed method is not clearly explained. Different concepts are discussed, but there is no summary and discussion of the proposed method as a whole. The reader must infer how the method works from the different pieces.

When discussing the performance of different methods, and even if in the text the four different alternatives are clearly explained, in figure captions and legens the terminology changes (ISA-VAE, ISA-beta-VAE, beta-VAE, beta-ISA-VAE, etc). This makes the discussion very difficult to follow, as we do not understand which figures are comparable to which, and in which respect.

In addition, there are other (secondary) questions that require an answer.
S1) After (10) you mention the subspaces v_1,...v_l_o. What is the formal definition of these subspaces?
S2) The definition of the distribution associated to ISA also implies that n_i,k = 1 for all i and k, right?
S3) Could you please formally write the family of distributions, since applying this to a VAE is the main contribution of your manuscript?
S4) Which parameters of this family are learned, and which of them are set in advance?
S5) From Figure 4 and 5, I understand that the distributions used are of the type in (7) and not (10). Can you comment on this?
S6) How is the Lp layout chosen?
S7) Why the Lp layout for ISA-beta-VAE in Figure 5 is not the same as in Figure 4 for ISA-VAE?
S8) What are the plots in Figure 4? They are difficult to interpret and not very well discussed.

Finally, there are a number of minor corrections to be made.
Abstract: latenT
Equation (3) missig a sum over j
Figure 1 has no caption
In (8), should be f(z) and not x.
Before (10), I understand you mean Lp-nested
I did not find any reference to Figure 3
In 4.1, the standard prior and the proposed prior should be referred to with different notations.

For all these reasons I recommend to reject the paper, since in my opinion it is not mature enough for publication.

---

> ### Author Response · Authors · 2018-11-26
> **Comment to Reviewer 2**
>
> We thank Reviewer 2 for the constructive feedback and appreciate that R3 "agree[s] with the motivation of the manuscript, and in particular the choice of the prior distribution". Also we thank Reviewer 3 for further comments that helped to improve the clarity of the paper.
>
> We are glad that both reviewers R2 and R3 support the general approach taken and found the motivation of our work and the experiments convincing. We believe that the revised manuscript addresses all of the concerns of R3. Sincerely we hope that R3 might want to reconsider the revised manuscript for publication when coming to the final review score at the end of the rebuttal phase.
>
> We will now reply to the concerns raised by R3 one by one:
>
> P1) As in the original VAE approach, the form of the approximate posterior q(x) is a Gaussian distribution, for which mean and variance are defined by the encoder. We now mention this explicitely in the manuscript.
>
> P2) We agree with Reviewer 3 that the reparameterization trick is crucial for the variational autoencoder approach. We'd like to point out that the proposed approach is fully compatible with the reparameterization trick:
>
> To use the proposed prior distribution in a variational autoencoder, the only requirement is that we are able to compute the log density log p_ISA(z) of a sample z. The density is defined in Eq. 7.
>
> The KL-divergence can then be computed for each sample z by
>
> - log p_ISA(z) + log q_Gaussian(z|x)
>
> As discussed in Roeder et al. 2017 this approach even has potential advantages (variance reduction) in comparison to a closed form KL-divergence. We discuss this further at the end of Sec. 3.2.
>
> We do not have to modify the approximate posterior, thus the reparameterization trick can be applied. This is now described in the paragraph "Sampling and the Reparameterization Trick".
>
> If we also want to sample from the trained generative model, the other requirement is that we are able to sample from the prior distribution.
> This is indeed possible for the proposed prior (Sinz and Bethge 2010), and we include the sampling scheme in the appendix as Algorithm 1.
>
>
> P3) To choose the layout we follow a strategy similar to the one proposed in Sinz et al. 2009b and evaluate the MIG scores and reconstruction loss for different layouts. The best performing layouts are compared in Fig. 6. To increase the clarity of the plot we now show mean values with error bars.
>
>
> Description of the overall method:
>
> We added a detailed description of the modified ELBO for ISA-VAE and ISA-TCVAE at the end of Sec. 3.2.
>
>
> Use of the terms ISA-VAE, ISA-beta-VAE, beta-VAE, beta_ISA-VAE, etc.
>
> We thank the reviewer for this comment as it helps to enhance the readability of the manuscript. We have simplified the terms that denote instances of the proposed method to ISA-VAE and ISA-TCVAE. Please note that the terms "beta-VAE" and "beta-TCVAE" were defined in their respective publications and we will continue to use these terms to denote these approaches.
>
> S1) We revisited the original definitions in Sinz and Bethge 2010 and found an inconsistency: v_1, ..., v_l_0 were used to denote both the function values (Table 1 in Sinz and Bethge 2010) and the subspaces themselves (p. 3433 in Sinz and Bethge 2010). We now denote the function values with v_1, ..., v_l_0 and the subspaces with \mathcal(V)_1, ..., \mathcal(V)_l_0 and provide definitions of both in the revised manuscript. Thank you for drawing our attention to this.
>
> S2) This statement holds for the children of the root node i in 1,...,l_0.
> For the root node i=0 it holds that
> n_{0,k} = n_{k} for k in 1,...,l_0
> ie. the dimensionality of each ISA subspace. Please also refer to Fig. 1 (b) which visualizes an ISA model.
>
> S3) The Lp-nested distribution is defined in Eq. 7, and the family of ISA-models is obtained by plugging in Lp-nested functions of the form of Eq. 9 into Eq. 7. To increase clarity we now denote the probability density Eq. 7 with p_ISA(z) instead of rho(z).
>
> S4) The only free parameters of the family are the parameters of the ISA-layout and those are evaluated as described in our response to P3. We think that learning these parameters is an exciting direction for future research.
>
> S5) Eq. 7 [now Eq. 6] is an example of an Lp-nested distribution and is presented for didactic purposes only. The ISA models used in the experiments use the subclass of Lp-nested distributions as defined in Eq. 10 [now Eq. 9].
>
> S6) See response to S4 and P3.
>
> S7) The result presented in Fig. 4 a) is a result from an early experiment. We now present the result for the same parameters as in Fig. 5.
>
> S8) These plots are the standard evaluation plots for the dSprites dataset as introduced in Chen et al. 2018. We added a description to the figure caption and the appendix.
>
>
> Roeder, G., Yuhaei, W., and Duvenaud. D., Sticking the Landing: Simple, Lower-Variance Gradient Estimators for Variational Inference, NIPS 2017

---

### Official Review · AnonReviewer3 · 2018-11-05
**Several Interesting New Priors Are Proposed For The Latent Variables in VAE**

**Rating:** 7
**Confidence:** 4

**Review:**

The authors point out several issues in current VAE approaches, including the rotational symmetric Gaussian prior commonly used. A new perspective on the tradeoff between reconstruction and orthogonalization is provided for VAE, beta-VAE, and beta-TCVAE. By introducing several non rotational-invariant priors, the latent variables' dimensions are more interpretable and disentangled. Competitive quantitative experiment results and promising qualitative results are provided. Overall, I think this paper has proposed some new ideas for the VAE models, which is quite important and should be considered for publication.

Here I have some suggestions and I think the authors should be able to resolve these issues in a revision before the final submission:
1) The authors should describe how the new priors proposed work with the "reparameterization trick".
2) The authors should at least provide the necessary implementation details in the appendix, the current manuscript doesn't seem to contain enough information on the models' details.
3) The description on the experiments and results should be more clear, currently some aspects of the figures may not be easily understood and need some imagination.
4) There are some minor mistakes in both the text and the equations, and there are also some inconsistency in the notations.

---

> ### Author Response · Authors · 2018-11-26
> **Comment to Reviewer 3**
>
> We are glad that R3 appreciates the "competitive quantitative experiment results and promising qualitative results" and finds an important contribution to the state-of-the art in our publication.
> We also thank for the constructive feedback which we address in the following:
>
> 1) Since we only modify the prior and keep the approximate posterior following the original VAE approach as a multivariate Gaussian, the reparameterization trick can be applied as usual. The proposed method only requires to compute the log likelihood of the prior. We added a paragraph to section 3 that explains this in more detail.
>
> 2) We added a description of the training process and the code of the encoder and decoder to the appendix.
>
> 3) We reworked the figures and captions and provide more details on the experiments especially towards reproducability and clarity.
>
> 4) We have reworked the notations towards correctness and clarity.

---

### Official Review · AnonReviewer4 · 2018-11-11
**The paper was not clearly written and failed to provide enough details.**

**Rating:** 4
**Confidence:** 3

**Review:**

The paper used the family of $L^p$-nested distributions as the prior for the code vector of VAE and demonstrate a higher MIG. The idea is adopted from independent component analysis that uses rotationally asymmetric distributions. The approach is a sort of general framework that can be combined with existing VAE models by replacing the prior. However, I think the paper can be much improved in terms of clarity and completeness.

1. The authors used MIG throughout section 4. But I have no idea what it is. Does a better MIG necessarily imply a good reconstruction? I am not sure if we can quantify the model performance by the mere MIG, and suggest the authors provide results of image generations as other GAN or VAE papers do.
2. Is the "interpretation" important for high dimensional code $z$? If yes, can the authors show an example of interpretable $z$?
3. I had difficulty reading Section 4, since the authors didn't give many technical details; I don't know what the encoder, the decoder, and the specific prior are.
4. The authors should have provided a detailed explanation of what the figures are doing and explain what the figures show. I was unable to understand the contribution without explanations.
5. Can the authors compare the proposed prior with VampPrior [1]?

The paper should have been written more clearly before submission.
[1] Tomczak, Jakub M., and Max Welling. "VAE with a VampPrior." arXiv preprint arXiv:1705.07120 (2017).

---

> ### Author Response · Authors · 2018-11-26
> **Comments to Reviewer 4**
>
> We thank the reviewer for the constructive feedback that improved the clarity of the paper. We have reworked section 4 and improved the figure captions.
>
> 1.  "The authors used MIG throughout section 4. But I have no idea what it is."
>
> We extended our description of the MIG score in section 4.
>
> "Does a better MIG necessarily imply a good reconstruction?"
>
> To evaluate the quality of the reconstruction we reported the log likelihood of the reconstruction in Fig. 3, 5 and 6. We demonstrate with our quantitative evaluation that the proposed prior mitigates the trade-off between reconstruction quality and disentanglement (this trade-off is discussed in section 4.3)
>
> "I am not sure if we can quantify the model performance by the mere MIG, and suggest the authors provide results of image generations as other GAN or VAE papers do."
>
> As requested we provide several results of image generations: latent traversals in the main manuscript in Fig. 3 and in the appendix in Fig. 7, 8 and 9. We also provide examples of image reconstruction in Fig. 10 in the appendix.
> As remark on this topic: We preferred a quantitative evaluation over a qualitative demonstration. Our quantitative analysis uses the data of 16 evaluations * 11 different values of beta = 176 trained models for each method, meaning that Fig. 4 aggregates the results of 704 experiments. We believe that this provides more evidence than a qualitative demonstration of generated images from a single successful experiment.
>
>
> 2. "Is the 'interpretation' important for high dimensional code $z$? If yes, can the authors show an example of interpretable $z$?"
>
> To provide an example of an interpretable z we depicted the standard evaluation plots of the MIG score in Fig. 4, that are produced with the reference implementation of Chen et al. 2018 available on https://github.com/rtqichen/beta-tcvae
> In fact the different dimensions of z encode individual underlying generative factors of the dataset, namely position in x, position in y, scale, and rotation angle. We have added these interpretations of the latent dimensions to Fig. 3.
>
>
> 3. I had difficulty reading Section 4, since the authors didn't give many technical details; I don't know what the encoder, the decoder, and the specific prior are.
>
> The prior is the independent subspace analysis model that is proposed in this paper.
> We used the same encoder and decoder architecture as in Chen et al. 2018 and have added a detailled description to the appendix A.5.
>
>
> 4. The authors should have provided a detailed explanation of what the figures are doing and explain what the figures show. I was unable to understand the contribution without explanations.
>
> We kindly ask the reviewer to specify which figures is referred to. We have reworked many of the figures and improved the captions. In Fig. 4 for example we follow standard practice established in Chen et al. 2018 for visualizing latent representations for the dSprites dataset.
>
>
> 5. Can the authors compare the proposed prior with VampPrior [1]?
>
> It would be interesting to consider a mixture of Gaussians model for learning latent representations.
> For the dataset we looked at specifically, a latent factor model is more appropriate (rather than a clustering model) which is captured by the proposed independent subspace model.

---

### Meta-Review · Area_Chair1 · 2018-12-11
**VAE with ISA prior**

**Confidence:** 5
**Recommendation:** Reject

**Metareview:**

The paper proposes to improve VAE by using a prior distribution that has been previously proposed for independent subspace analysis (ISA). The clarity of the paper could be improved by more clearly describing the proposed method and its implementation details. The originality is not that high, as the main change to VAE is replacing the usual isotropic Gaussian prior with an ISA prior. Moreover, the paper does not provide comparison to VAEs with other more sophisticated priors, such as the VampPrior, and it is unclear whether using the ISA prior makes it difficult to scale to high-dimensional observations. Therefore, it is difficult to evaluate the significance of ISA-VAE. The authors are encouraged to carefully revise their paper to address these concerns.